# 3D trajectories and velocities of rainfall drops in a multifractal turbulent wind field

Auguste Gires[1], Ioulia Tchiguirinskaia[1], and Daniel Schertzer[1]

[1]Hydrologie Meteorologie et Complexite (HM&Co), Ecole des Ponts Paris-Tech, Champs-sur-Marne, France

**Correspondence:** Auguste Gires (auguste.gires@enpc.fr)

**Abstract.** Weather radars measure rainfall in altitude whereas hydro-meteorologists are mainly interested in rainfall at ground level. During their fall, drops are advected by the wind which affects the location of the measured field.

The governing equation of a rain drop motion relates the acceleration to the forces of gravity and buoyancy along with the drag force. It depends non-linearly on the instantaneous relative velocity between the drop and the local wind; which yields to complex behaviour. Here, the drag force is expressed in a standard way with the help of a drag coefficient expressed as a function of Reynolds number. Corrections accounting for the oblateness of drops greater than 1-2 mm are suggested and validated through comparison of retrieved "terminal fall velocity" (i.e. without wind) with commonly used relationships in the literature.

An explicit numerical scheme then is implemented to solve this equation for 3+1D turbulent wind field, and hence analyse the temporal evolution of the velocities and trajectories of rain drops during their fall. It appears that multifractal features of the input wind are simply transferred to drop velocity with an additional fractional integration whose level depends on drop size, and a slight time shift. Using actual high resolution 3D sonic anemometer and a scale invariant approach to simulate realistic fluctuations of wind in space, trajectories of drop of various size falling form 1500 m are studied. For a strong wind event, drops located within a radar gate in altitude during 5 min are spread on the ground over an area of size few kilometers. Spread for drops of a given diameter are found to cover few radar pixels. Consequences on measurements of hydro-meteorological extremes which are needed to improve resilience of urban areas are discussed.

## 1 Introduction

During their fall, drops are advected by wind. Quantitative rainfall estimation with the help of weather radars are affected by this issue since drops can be displaced horizontally between their measurement location in altitude and their ground impact location which is of interest for hydro-meteorologists. This effect is usually called wind drift in the literature and sometimes wind advection. The potential bias and uncertainty introduced in radar measurement is stronger at higher resolution, i.e. typically with pixel size smaller than 1-2 $km^2$ which are needed for urban applications for example. Collier (1999) suggests that correction schemes should be implemented for this kind or higher radar resolution. Lauri et al. (2012) reported that far from radar (i.e. typically more than 150 $km$), even with low elevation (0.3°), displacements of few tens of $km$ are found, which actually distort the measured area.

Most correction schemes rely on the use of 4D wind profiles derived from numerical predictions models (Mittermaier et al., 2004; Lack and Fox, 2007; Lauri et al., 2012; Sandford, 2015) or combination of such with reanalysis (Dai et al., 2013, 2019; Yang et al., 2020). The latter also accounts for drop size distribution (DSD). With such, they report an improvement by $\approx 3\%$ of the correlation between radar and rain gauge measurement and a reduction of discrepancy of $\approx 18\%$ over eight selected events. Lack and Fox (2007) used directly Doppler radar wind measurement at 2.5 $km$ scale to adjust for wind drift effect. In general, correction schemes use wind data at rather coarse resolution (typically km(s)) and assume a constant wind shear. Nevertheless, some variability at smaller space-time scales is usually acknowledged, especially during convective events, i.e. the ones for which wind drift causes the greatest uncertainty (Lack and Fox, 2007).

Wind effects on rainfall drops is also reported to generate discrepancies between measured vertical velocities and expected terminal fall ones. For example, Montero-Martinez and Garcia-Garcia (2016) studied events with calm, light and moderate wind with various rainfall levels, and found a widening of the fall velocity distribution under windy conditions. They found super-terminal drops only for diameters $< 0.7$ $mm$ and more often under wind conditions. Sub-terminal fall velocities for drops of size up to 2 $mm$ are reported. Bringi et al. (2018) found that under low wind speed and turbulence, no discrepancies are found with expectations while under high wind speed and turbulence, there is a clear widening of the distribution. A linear decrease of mean fall velocity with increasing turbulent intensity is reported. Maximum decreases of 25–30 % are observed. Thurai et al. (2019) also found such decrease for drops greater than 2 $mm$ in high turbulence intensity conditions. It is associated to an asymmetry also appearing in the drop shape. They also found that drop horizontal velocities in both direction and magnitude show "remarkable agreement" with the wind sensor at 10 $m$. Stout et al. (1995) explored the effect of non linear drag coefficient on fall velocity through numerical simulations. They showed that even heavy drops exhibited a reduced settling velocity in isotropic turbulence.

Turbulence is found to have contradictory effects on the distribution of fall velocity. Indeed increasing turbulence level in windy and rainfall condition will yield to more collision and breakup, resulting in smaller drops inheriting the speed of larger parent ones, hence observations of super-terminal velocities. On the other end, turbulence is said to yield to a decrease in fall velocities because drops (especially ones $< 1$ $mm$) are more affected by eddies.

Such findings on the discrepancies between observed and expected fall velocities has effects on the relation between rainfall and kinetic energy, i.e. the erosivity 'power' of rainfall (Pedersen and Hasholt, 1995) and also building performance to outdoor conditions (Tian et al., 2018; Blocken et al., 2011).

The studies previously mentioned basically do not account for small scales wind fluctuations in both space and time. In this paper, we suggest to study the behaviour of individual rainfall drops of various sizes in a high resolution turbulent wind field. The variability of the wind is accounted for through the framework of Universal Multifractals (UM) (see Schertzer and Tchiguirinskaia, 2020, for a recent review). Such physically based framework is designed to analyse and simulate geophysical fields exhibiting extreme variability over wide range of space-time scales as wind. Drop oblateness is also accounted for.

The paper is organized as follow. In section 2, a deterministic equation for the fall of individual oblate drops in a 3D field is derived and validated through the comparison of obtained terminal fall velocity with commonly used formulas. In section 3, the framework of Universal Multifractals is briefly reminded. Then, the drops are subjected to simulated multifractal fields as

wind input and multifractal behaviour of horizontal drop velocity is assessed. Finally, in section 4, 3D wind is reconstructed from high resolution 3D sonic anemometer data and strong scaling assumptions. This field is used to study the trajectories of drops between falling from 1500 $m$ down to the ground.

## 2 A deterministic equation for oblate drops in a wind field

### 2.1 Formulation of the equation

Let us denote (x,y,z) the horizontal, lateral and vertical coordinates in a standard Cartesian framework with unit vectors ($\underline{e}_x$,$\underline{e}_y$,$\underline{e}_z$). We aim at writing the motion equation of a particle of water (a drop) of velocity $\underline{v}_p$, density $\rho_p$ and falling in the atmosphere under the influence of the gravity $\underline{g} = -g\underline{e}_z$ (where $g = 9.81$ $m.s^{-2}$) and the wind $\underline{v}_{wind}$ (with three components). The density of the atmosphere is denoted $\rho_{air}$. The water particle is characterized by its equivolumic diameter $D_{eq}$ which corresponds to the diameter of the sphere having the same total volume. Hence we have $Vol = \frac{\pi}{6}D_{eq}^3$. Finally the relative velocity between the wind and the falling particle is $\underline{v}_{rel} = \underline{v}_{wind} - \underline{v}_p$ ($\underline{v}_{rel}$ is the vector with three components while $v_{rel}$ is its euclidean norm).

The drop is subjected to three forces:

- The gravity equal to $\rho_p \, Vol\underline{g}$

- The buoyancy equal to $-\rho_{air} \, Vol\underline{g}$

- The drag, which is commonly written as $\frac{1}{2}\frac{\pi D^2}{4}c_D\rho_{air}v_{rel}\underline{v}_{rel}$. $Re$ is the common Reynolds number $Re = \frac{\rho_{air}v_{rel}D}{\mu_{air}}$ where $\mu_{air}$ is the absolute viscosity of air. $c_D$ is the drag coefficient and depends in general of $Re$ and $D_{eq}$. The next section is devoted to its determination.

As a consequence, the equation of motion of the falling particle is given by Newton's second law which equals the mass times the acceleration to the net force (here it was divided by the mass):

$$\frac{d\underline{v}_p}{dt} = \frac{3}{4D}c_D\frac{\rho_{air}}{\rho_p}v_{rel}\underline{v}_{rel} + \underline{g}\frac{\rho_p - \rho_{air}}{\rho_p} \tag{1}$$

### 2.2 Determination of the drag coefficient

Before discussing how the drag coefficient is determined, it should be reminded that rainfall drops which are considered in this paper are not spherical. Indeed drops greater than typically 1.5 $mm$ become oblate in their fall. This oblateness increases with size. A very commonly used model consists in an ellipsoid with an axis ratio varying depending on the size. Thurai et al. (2007) showed that such model is too simplistic since drops are not symmetric in the direction perpendicular to their fall. Following an in-depth analysis of the drop shape assessed with the help of a 2D-video disdrometer (Kruger and Krajewski, 2002) in the measurement campaign of an artificial rainfall experiment; they suggested the following formula for the shape:

$$x = a_1 \sqrt{1 - \left(\frac{z}{a_2}\right)^2} \left[cos^{-1}\left(\frac{z}{a_2 a_3}\right)\right] \left[a_4 \left(\frac{z}{a_2}\right)^2 + 1\right] = f(z) \tag{2}$$

with:

$$
\begin{aligned}
a_1 &= & \frac{1}{\pi}\left(0.02914 D_{eq}^2 + 0.9263 D_{eq} + 0.07791\right) \\
a_2 &= & -0.01938 D_{eq}^2 + 0.4698 D_{eq} + 0.09538 \\
a_3 &= & -0.06123 D_{eq}^3 + 1.3880 D_{eq}^2 - 10.41 D_{eq}^2 + 28.34 \\
a_4 &= & -0.01352 D_{eq}^3 + 0.2014 D_{eq}^2 - 0.8964 D_{eq}^2 + 1.226 \ if \ D_{eq} > 4 \ mm \\
a_4 &= & 0 \ if \ 1.5mm \le D_{eq} \le 4mm
\end{aligned} \tag{3}
$$

This shape corresponding to a solid of revolution around $z$ axis is used in this paper. It is displayed in Fig. 2.a for drops with equivolumic diameter ranging from 1.5 $mm$ to 5.5 $mm$. It should be mentioned that computing the volume as an integral of the shape ($Vol = \int_{z_{min}}^{z_{max}} \pi f(z)^2 dz$, see Fig. 1) yields to minor differences with the expected volume of $\frac{\pi D_{eq}^3}{6}$. They are highlighted

in Fig. 2.b. In order to account for this small difference, once an equivolumic diameter is set, the corresponding one that would lead to the expected volume from Eq. 2 and 3 is computed from a correspondence table. The relationship, which is obviously close to the bisector is displayed in Fig. 2.c with the horizontal axis corresponding to the real $D_{eq}$ of the drop and the vertical axis the $D_{eq}$ to be input in Eq. 2 and 3 to retrieve the correct expected volume. A consequence is that oblateness of drops will be considered only from equivolumic diameter greater than 1.527 $mm$.

For non spherical shapes, it is quite tricky to compute the corresponding drag coefficient as a function of the Reynolds number. The literature about this issue is quite abundant and the interested reader is referred to chapter 4 of the PhD dissertation of Baheri (2015) or Hölzer and Sommerfeld (2008) for details. In the approach they implemented, three parameters are used to characterize the non spherical shapes of the falling particle with the help of three dimensionless parameters: The sphericity, the crosswise sphericity and the lengthwise sphericity. The two last depend on the orientation of the particle with regards to the

flow. Here it is assumed that drops are oriented perpendicularly to the flow, i.e. the 'z' axis of Eq. 2 is parallel to $\underline{v}_{rel}$. In the general case, these parameters may be complex to assess but with the shape derived from Eq. 2 (Thurai et al., 2007), theoretical formula can be obtained. See Fig. 1 for an illustration of the computations via integral calculus. The three parameters are:

– The sphericity $\psi$ which is equal to ratio between the surface area ($SA$) of the equivolumic sphere to the actual surface area of the particle. It is equal to one for sphere and decreases for less and less spherical particles. $\psi = \frac{pi D_{eq}^2}{Surface\ Area}$. In

the framework of this paper, i.e. we have $SA = \int_{z_{min}}^{z_{max}} 2\pi f(z)\sqrt{1 + f'(z)^2} dz$

– The crosswise sphericity $\psi_\perp$ which is equal to ratio between the projected area of the volume equivalent sphere and the projected area of the particle normal to the falling direction (here $\underline{e}_z$). $\psi_\perp = \frac{D_{eq}^2}{D_{max}^2}$. It is equal to one for sphere and decreases for larger drops since they become oblate.

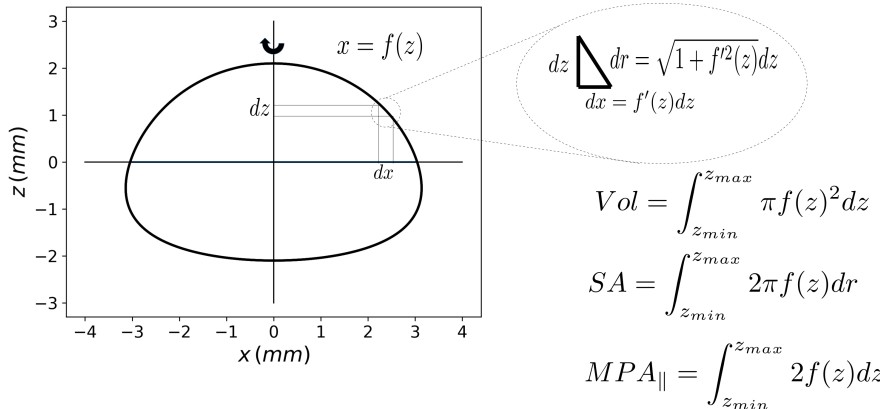

$$Vol = \int_{z_{min}}^{z_{max}} \pi f(z)^2 dz$$

$$SA = \int_{z_{min}}^{z_{max}} 2\pi f(z) dr$$

$$MPA_\parallel = \int_{z_{min}}^{z_{max}} 2f(z) dz$$

**Figure 1.** Illustration on how the volume ($Vol$), the surface area ($SA$), and the mean projected longitudinal cross-sectional area ($MPA_\parallel$) of the particle used in this paper can be computed via integral calculus. The drop is actually a solid of revolution around the 'z' axis.

– The lengthwise sphericity $\psi_\parallel$ which is defined as to the cross-sectional area of the volume equivalent sphere divided by the difference between half the surface area and the mean projected longitudinal cross-sectional area of particle ($MPA_\parallel$). $\psi_\parallel = \frac{\pi(\frac{D_{eq}}{2})^2}{\frac{SA}{2} - MPA_\parallel}$. In the specific drop model of this paper, we have $MPA_\parallel = \int_{z_{min}}^{z_{max}} 2f(z) dz$ (it basically corresponds to the area in 2D of the drop plotted in Fig. 1).

The evolution of these parameters as a function of $D_{eq}$ for the considered drops is in Fig. 2.d. The increasing oblateness of drops with increasing size is translated through the fact that the parameters are getting further away from 1. In order to define the drag coefficient, the corrections suggested by Hölzer and Sommerfeld (2008) to account for non sphericity of particles are then implemented on the formula of White (1974) previously used by Stout et al. (1995) who worked only on spherical drops. This yields:

$$c_D = \frac{8}{Re\sqrt{\psi_\parallel}} + \frac{16}{Re\sqrt{\psi_\perp}} + \frac{6}{(1+\sqrt{Re})\psi^{3/4}} + \frac{0.25 \times 10^{0.4(-log\psi)^{0.2}}}{\psi_\perp} \tag{4}$$

The evolution of $c_D$ as a function of $Re$ for various drop parameters is displayed in Fig. 2.e and follows standard patterns.

## 2.3 Validation of the formula

In order to validate the developed equation, the retrieved terminal fall velocity is assessed for each equivolumic diameter. It corresponds to the velocity of the permanent regime with no wind, i.e. the drag plus the buoyancy exactly compensate the gravity. Computations are carried out with $\rho_{air} = 1.205 \ kg.m^{-3}$, $\mu_{air} = 1.81 \ 10^{-5} \ kg.m^{-1}.m{-2}$, $\rho_{water} = 998.2 \ kg.m^{-3}$ $g = 9.81 \ m.s^{-2}$ as in Stout et al. (1995).

The relation between obtained terminal fall velocity vs. equivolumic diameter is displayed in red in Fig. 2.f. The developed equations enables to retrieve commonly used relation (Beard, 1977; Lhermitte, 1988; Best, 1950; Atlas et al., 1973) for drops

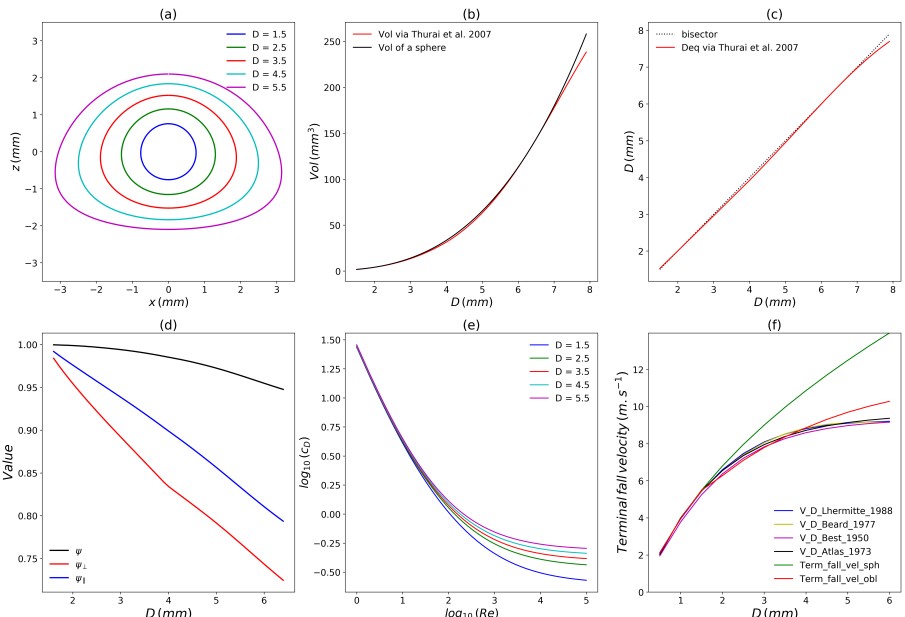

**Figure 2.** (a) Drop shape model used in this paper; (b) Drop volume vs. equivolumic diameter; (c) Diameter relation to retrieved wanted volume; (d) Parameters characterizing non spherical shape of drops vs. equivolumic diameter; (e) Drag coefficient $c_D$ vs. $Re$ number; (f) Terminal fall velocity vs. equivolumic diameter

of diameter up to 4 $mm$. The deviations found when considering spherical drops (in green) are visible for diameter greater than 2 $mm$ which highlights the need to account for drop oblateness.

### 2.4 Numerical scheme for solving the equation

Eq. 1 is solved numerically through the implementation of a simple Eulerian numerical scheme. In such framework: (i) a discretisation of time with time step $\Delta t$ is introduced yielding to discrete time steps $t_n = n \times \Delta t$ where $n$ is an integer; (ii) we aim at finding an approximation of $\underline{v}_p$ at time step $n$ denoted $\underline{v}_{p,n}$; (iii) the first derivative in Eq. 1 is approximated as $\frac{d\underline{v}_p}{dt}(t_n) \approx \frac{\underline{v}_{p,n+1} - \underline{v}_{p,n}}{\Delta t}$. This yields to the following equation for the numerical scheme:

$$\underline{v}_{p,n+1} = \underline{v}_{p,n} + \Delta t \left[ \frac{3}{4D} c_{D,n} \frac{\rho_{air}}{\rho_p} v_{rel,n} \underline{v}_{rel,n} + \underline{g} \frac{\rho_p - \rho_{air}}{\rho_p} \right] \tag{5}$$

where $\underline{v}_{rel,n}$ and $c_{D,n}$ are computed at time step $t_n$ using the formulas discussed in section 2.1 and 2.2. Assuming some initial conditions (always no horizontal velocity and a vertical one equal to the terminal fall for the corresponding diameter), it is then possible to reconstruct the time series of velocity for the drops. From it, the temporal evolution of the position (i.e. the trajectory) is derived. It is needed to properly assess the wind accounting for the current position of the drop. A time step of $\Delta t = 0.01\ s$ is used in this paper, and it was checked that it ensured a stability of the numerical scheme.

## 3 Behaviour of horizontal drop velocity with multifractal input

### 3.1 Brief reminder on Universal Multifractal framework

It is outside the scope of the paper to introduce in details the framework of Universal Multifractals (UM). Hence, only the most important elements are reminded here and interested readers are referred to the references mentioned or to a recent review by Schertzer and Tchiguirinskaia Schertzer and Tchiguirinskaia (2020) for more details.

Let us consider a field $\epsilon_\lambda$ at a resolution $\lambda$ defined as the ratio between the outer scale ($L$) and observation scale ($l$); $\lambda = L/l$. For multifractal fields, the moment of order $q$ of the field is power law related to the resolution:

$$\langle \epsilon_\lambda^q \rangle \approx \lambda^{K(q)} \tag{6}$$

where $K(q)$ is the scaling moment function. It fully characterizes the variability across scales of the field. In the specific framework of UM (Schertzer and Lovejoy, 1987, 1997), towards which multiplicative cascades processes converge, only two parameters with physical interpretation are needed to characterize $K(q)$ for conservative fields:

- $C_1$, the mean intermittency co-dimension, which measures the clustering of the (average) intensity at smaller and smaller scales. $C_1 = 0$ for an homogeneous field;

- $\alpha$, the multifractality index ($0 \leq \alpha \leq 2$), which measures the clustering variability with regards to the intensity level.

For UM, we have:

$$K(q) = \frac{C_1}{\alpha - 1}(q^\alpha - q) \tag{7}$$

A non-conservative field ($\psi_\lambda$), i.e. whose mean is not preserved across scale can be written as $\psi_\lambda \approx \epsilon_\lambda \lambda^{-H}$, where $H$ is the non-conservativeness parameter. $H = 0$ for conservative fields. Positive values correspond to a fractional integration to go from $\epsilon_\lambda$ to $\psi_\lambda$ and to stronger correlations within the field $\psi_\lambda$. Negative values correspond to a fractional differentiation. $H$ is typically between 0 and 1 for geophysical fields.

The first step of a multifractal analysis usually consists in a spectral analysis. For multifractal fields the power spectra ($E$) should scale with wave number $k$:

$$E(k) = k^{-\beta} \tag{8}$$

with the spectral slope $\beta$

$$\beta = 1 + 2H - K_c(2) \tag{9}$$

where $K_c$ is the scaling moment function (Eq. 7) of the conservative part of the field. To analyse the latter, a Trace Moment (TM) is implemented. It notably enables to assess the quality of the scaling behaviour. It basically consists in plotting Eq. 6

in log-log. Straight lines should be retrieved and the slope gives $K(q)$. Finally, UM parameters are estimated with the help of the Double Trace Moment (DTM) technique which is tailored for UM fields and enables robust estimation of UM parameters (Lavallée et al., 1993).

## 3.2 Methodology

In this section, the scaling behaviour of horizontal drop velocity is assessed using numerical simulations. Working with such input whose features are fully known is helpful to understand how drops react to wind.

More precisely, a horizontal input $v_{x,wind}$ for Eq. 1 is simulated with the help of blunt multifractal discrete cascades (Gires et al., 2020). Such process yields only positive values which is not realistic for wind. Hence a standard 'complex trick' was used to generate a field with both positive and negative values (Schertzer and Lovejoy, 1995). To implement it, two fields $X_1$ and $X_2$ are generated with the wanted features, and a third one is obtained with the help of the following equation ($Real$ is the real part):

$$X = Real \left[ exp \left( \log X_1 + i \log X_2 \right) \right] \tag{10}$$

Such field divided by two was used as input. 1024 time step long series are generated with UM parameter $\alpha = 1.7$ and $C_1 = 0.2$, which corresponds to typical value for turbulent wind fields (Fitton et al., 2011). The time step is assumed to be of 0.01 $s$ which means that drops are basically studied over 10 $s$. For the initial conditions, drops are assumed to have no horizontal velocity and a vertical component equal to its corresponding terminal fall velocity. Since scaling is a statistical behaviour, an ensemble of 100 independent samples was generated and the corresponding ensemble of horizontal drop velocity was simulated using Eq. 5 for drops of various sized ($D_{eq} \subset [0.1, 0.2, 0.4, 0.6, 0.8, 1, 1.5, 2, 3, 4]$)

## 3.3 Results and discussion

Fig. 3 displays the temporal evolution of drops' horizontal velocity over 10 $s$ for a sample of wind input (in black). Three drop diameters are displayed (0.1, 0.6 and 2 $mm$). It can be seen, notably on the zoomed part of the figure (lower panel) that the smaller drop ($D_{eq} = 0.1 \ mm$, in blue) follows well wind fluctuations with only a limited dampening of the fluctuations. A small delay ($\approx 0.01 \ s$) corresponding to a reaction time is noted. As it can be expected larger drops ($D_{eq} = 0.6 \ mm$, in green; and $D_{eq} = 2 \ mm$, in red) tend to dampen even more wind fluctuations.

In order to quantify more precisely this qualitative behaviour, a multifractal analysis on the retrieved ensembles was performed. Fig. 4 displays the outcome of spectral and TM analysis for drops of equivolumic diameter ranging from 0.1 to 2 $mm$. The spectral analysis reflect a good scaling behaviour over the whole range of scales. Spectral slopes ($\beta$ in Eq. 8) of 0.86 and 2.25 are retrieved respectively. For the 2 $mm$ drop, the value corresponds to non-conservative fields. In order to ensure that a conservative field is studied in TM analysis, which is needed (Lavallée et al., 1993), a fractional differentiation with an exponent $(\beta - 1)/2$ is implemented on the field before implementing this TM analysis. TM analysis is displayed in the right column of Fig. 4. For the 0.1 $mm$ drop, an excellent scaling behaviour is retrieved with coefficient of determination $r^2$ for $q = 1.5$ greater than 0.99. DTM analysis yields estimates of UM parameters $\alpha$, $C_1$ and $H$ equal to 1.68, 0.21 and 0.12

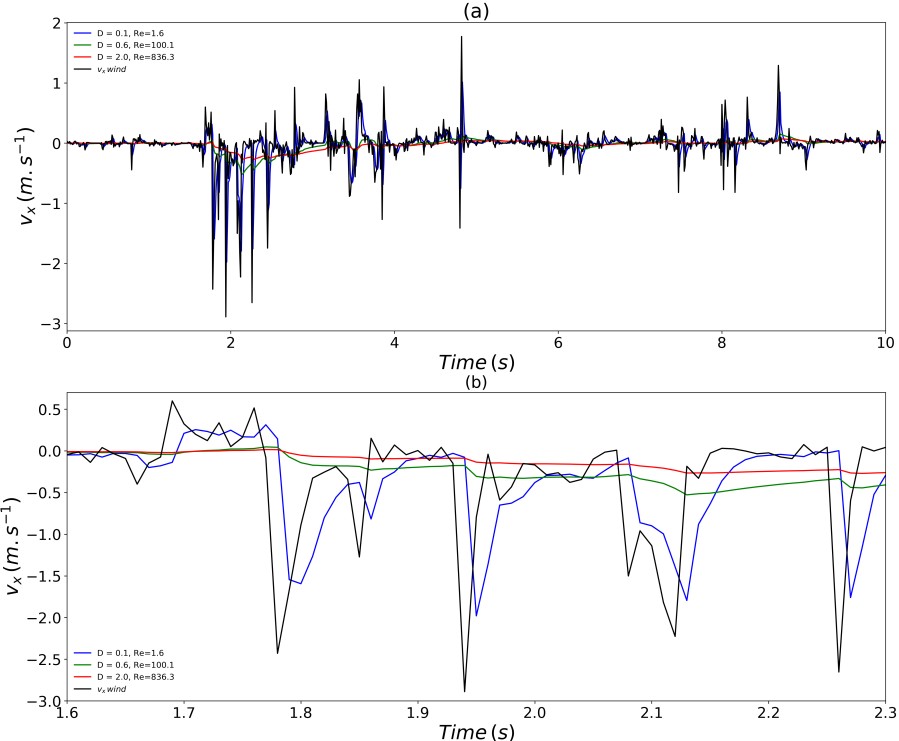

**Figure 3.** Top: temporal evolution of the drop horizontal velocity during 10 $s$ for various drop diameters (it is the equivolumic diameter that is indicated) with the same multifractal wind input (in black). Bottom: a zoom of the above curve on a shorter period

respectively, which is close to the features of the input series. For the 2 $mm$ drop, the scaling is slightly degraded but remains good ($r^2 = 0.95$ for $q = 1.5$) and we find $\alpha = 1.69$, $C_1 = 0.14$ and $H = 0.79$.

Fig. 5 displays a summary of the UM analysis carried out on the generated series for the various drops. The scaling behaviour is excellent for small drops and remains good for all drop sizes with $r^2$ for $q = 1.5$ always greater than 0.95 (Fig. 5.e). The need for a fractional differentiation before implementing TM analysis is visible with the very poor scaling found when analysing directly the field. The non-conservativeness parameter rapidly increases from 0.1 to 0.8 with drop size increasing from 0.1 $mm$ to $\approx 1 - 1.5$ $mm$. For larger drops it remains rather stable. This increase of $H$ is basically a quantification of the increased dampening of wind fluctuations observed for larger drops discussed with Fig. 3. With regards to the UM parameters $\alpha$ and $C_1$, the former remains stable and close to the input value of 1.7 for all drop size. The latter exhibits a small decrease with larger drops. It should be reminded that this approach is somehow artificial since all drops are perceiving the same wind, which would not be the case in reality because they do not fall at the same vertical speed. In summary, this investigation shows that horizontal drop velocity basically reproduces the multifractal properties of the wind input with an increased level of non-conservativeness $H$. $H$ strongly increases for drops smaller than 1 $mm$ and then stabilizes.

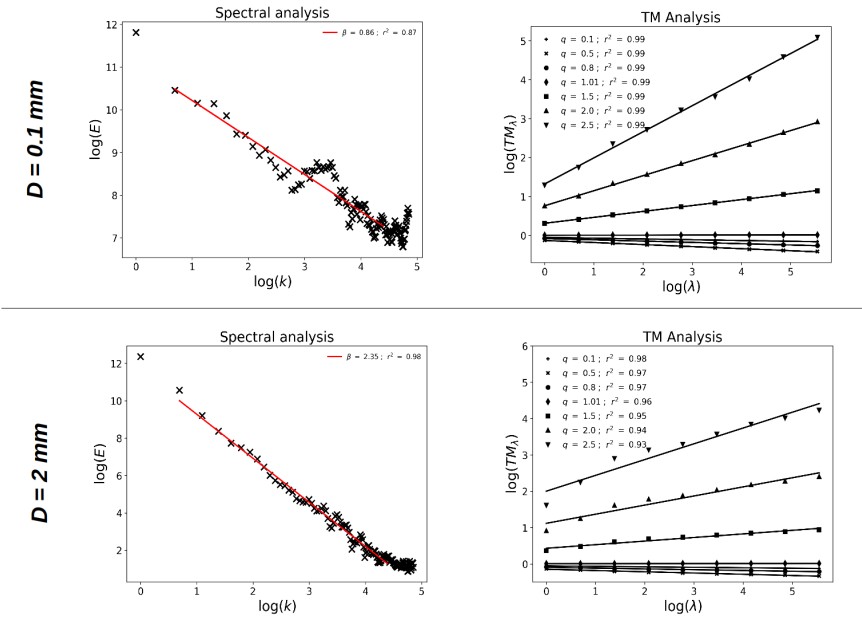

**Figure 4.** Scaling behaviour of the simulated drop velocity for $D_{eq}$ equal to 0.1 $mm$ (top row) and 2 $mm$ (bottom row). Spectral analysis (i.e. Eq. 8 in log-log) on the direct simulations is displayed in the left column. TM analysis (i.e. eq. 6 in log-log) on the velocity after implementing a fractional integration is displayed in the right column

## 4 Ground impact location of drops falling in a turbulent wind field

### 4.1 Methodology

The purpose of this section is to investigate where drops falling from a height of 1500 $m$ reach the ground. Given the time
step of 0.01 $s$ used in the equation and the fact that drops are moving in space during their fall, it means that having high resolution space-time 3D wind data over an area of typical size few kilometers is needed to fully address the issue. Such data is unfortunately not available. Hence we suggest here to reconstruct a somehow realistic wind from a punctual measurement relying on previous findings on turbulence.

#### 4.1.1 Wind data

More precisely, we use 100 Hz 3D sonic anemometer data collected at by a device installed at 78 $m$ on meteorological mast located on the Pays d'Othe wind farm in the framework of the ANR RW-Turb project. The wind farm is at roughly 120 $km$ South-East of Paris on a slightly sloppy area. More details can be found in the data paper under discussion at ESSD (Gires et al., 2022) and in the data set (Gires et al., 2021). Two wind series with very different average horizontal wind speed (i.e. 1.8 vs. 11.8 $m.s^{-1}$) were extracted for this study. They are later denoted low wind event and strong wind event. The corresponding

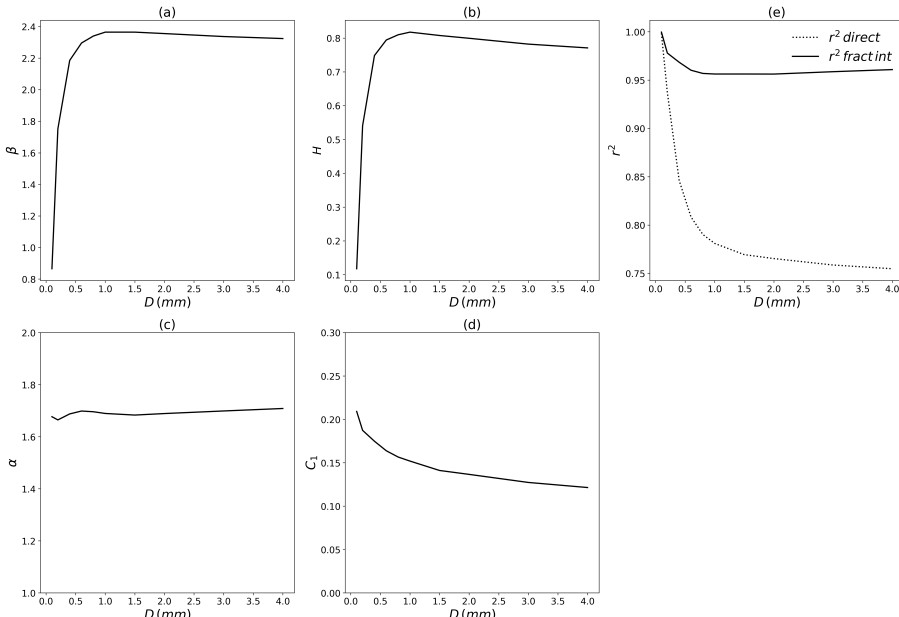

**Figure 5.** Summary of the multifractal analysis performed on the ensembles of simulated drop horizontal velocities using a wind input with $\alpha = 1.7$, $C_1 = 0.2$. The various multifractal parameters are displayed vs. $D$ (the equivolumic drop diameter)

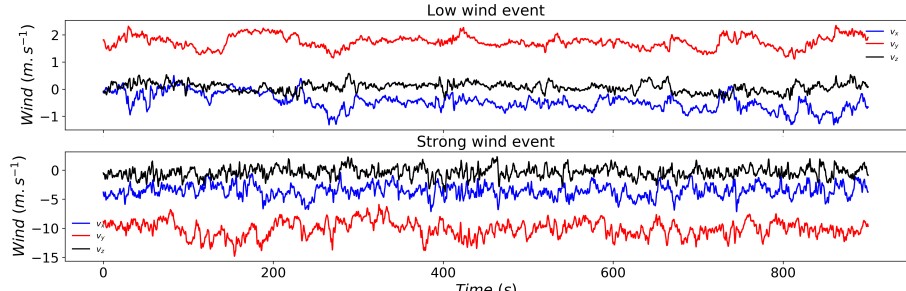

**Figure 6.** Temporal evolution of the 100 Hz wind data from 3D sonic anemometer for low (top) and strong (bottom) wind event used in this paper

time series of the three components of the wind for both events are displayed in Fig. 6 over approx. 900 $s$. The low wind event was collected on 20/01/2021 while the strong one occurred on 06/01/2021.

### 4.1.2 Generation of an anisotropic 3D turbulent field

In this section we discuss how to stochastically generate a turbulent field reproducing as well as possible the physics of such a flow constraint to have the empirical velocity values $v(x, y, z, t)$ of the 3D sonic anemometer located at $(x, y, z, t)$.

*Scaling and anisotropy*

It is quite obvious that gravity has such a strong impact on drop trajectories and dynamics of drops that classical scaling approaches just fail because they presuppose isotropy. On the contrary, the anisotropy between the vertical and the horizontal induced by gravity is so ubiquitous in geophysics that it has led to the general concept and framework of 'generalised scale invariance' for the analysis and simulation of anisotropic fields (Schertzer and Lovejoy, 1985, 1988, 1989; Lazarev et al., 1994; Schertzer et al., 2012) and new extensions have been developed for vector fields (Schertzer and Tchiguirinskaia, 2015, 2020). While classical approaches consider scaling only after assuming isotropy, general scale invariance first posit scaling and then study the remaining non trivial symmetries. Because this paper deals with scalar discrete cascades, and therefore their limitations, we consider an oversimplification of the generalised scale invariance. Recall that for the simplest case of generalised scale invariance, a $v$ horizontal component of a statistically translation invariant velocity field satisfies the following self-affine scale invariance:

$$\Delta v(T_\lambda \Delta \underline{x}) \overset{d}{=} \lambda^{H_h} \Delta v(\Delta \underline{x}) \tag{11}$$

where $\overset{d}{=}$ stands for equality in distribution, $\Delta \underline{x}$ for a (vector) pair separation, $\Delta v = |v(\underline{x} + \Delta \underline{x}) - v(\underline{x})|$ for the induced velocity component shear, $H_h$ for the horizontal scaling exponent ($H_h = 1/3$ for Kolmogorov's scaling) and $T_\lambda$ for a self-affine change of scale:

$$T_\lambda = \lambda^G = \exp(\log(\lambda G)); G = \begin{pmatrix} 1 & 0 & 0 \\ 0 & 1 & 0 \\ 0 & 0 & H_z \end{pmatrix} \tag{12}$$

where $G$ is the generator of $T_\lambda$, $H_z = H_h/H_v$ is the scaling anisotropy exponent, $H_v$ the scaling exponent along the vertical. $H_v = 3/5$ for a Bolgiano-Obukhov scaling along the vertical and therefore $H_z = 5/9$, while for isotropic 3D turbulence $H_z = 1, H_h = H_v$. The main property of $T_\lambda$ is that it broadly generalises in a straightforward manner scales $|.|$ from classical isotropic frameworks ($H_z = 1$) to generalised scales $\|.\|$ of anisotropic frameworks (e.g., $H_z \neq 1$), so that any isotropic cascade can be transformed in this manner into an anisotropic cascade without anything else. Just recall the canonical example of generalised scales for a diagonal generator ($p \geq 1$):

$$G_{i,j} = g_i \delta_{i,j} : \|\underline{x}\|_p = (\sum_i |x_i|^{p/g_i})^{1/p} \Rightarrow \|T_\lambda \underline{x}\|_p = \lambda \|\underline{x}\|_p \tag{13}$$

which displays the (generalised) scaling property of the generalised scales.

Equation 11 is valid for any direction of $\Delta \underline{x}$ ( $\underline{x} = (x, y, z)$), in particular along the horizontal $\Delta x$ and the vertical $\Delta z$:

$$\Delta v(\lambda \Delta x) \overset{d}{=} \lambda^{H_h} \Delta v(\Delta x); \Delta v(\lambda \Delta z) \overset{d}{=} \lambda^{H_v} \Delta v(\Delta v) \tag{14}$$

which can be respectively interpreted like:

$$\Delta v(\Delta x) \overset{d}{=} \varepsilon(\Delta x)^{a_h} \Delta x^{H_h}; \Delta v(\Delta z) \overset{d}{=} \phi(\Delta z)^{a_v} \Delta z^{H_v} \tag{15}$$

with scaling exponents $a_h = 1/3$ for the kinetic energy flux density $\varepsilon$ and $a_v = 1/5$ for the buoyancy force flux density $\phi$. Due to the fact that eddies can be defined as structures having a given velocity shear $\Delta v$, we have in fact the following balance between both flux densities:

$$\varepsilon (\Delta x)^{a_h} \Delta x^{H_h} \overset{d}{=} \phi (\Delta z)^{a_v} \Delta z^{H_v} \tag{16}$$

and finally:

$$\varepsilon (\Delta x)^{a_h} \overset{d}{=} \phi (\Delta z)^{a_v}, \text{ with} : \Delta x^{H_h} \approx \Delta z^{H_v} \tag{17}$$

due to the respective scaling of the horizontal and vertical eddy sizes defined by the scale change $T_\lambda$ (Eq.12).

This confirms that $\phi$ and $\varepsilon$ are both sites of the same coin (Eq.11). In summary, what is only needed is an anisotropic cascade defined by an anisotropic scale compatible (e.g., Eq.14) with the anisotropic scale change $T_\lambda$ (Eq.12), the intermittency being introduced either by the flux density $\varepsilon$ or $\phi$ (Eq. 15), which respect their equivalence (Eq. 17).

Practical difficulties only occur with discrete scales because they cannot easily deal with arbitrary $H_z$.

*Discrete scales and an ad-hoc approximation*

To get around these difficulties, it was tentatively proposed (REF EGU conf.?) to consider the following 3D model:

$$
\begin{aligned}
v_x(x+\Delta x, y+\Delta y, z+\Delta z, t) &= v_x(x,y,z,t) + c_{\epsilon_x}\, \epsilon_x(x+\Delta x, y+\Delta y, t)^{a_h} \Delta l^{H_h} + c_{\theta_x}\, \theta_x(z+\Delta z, t)^{a_v} \Delta z^{H_v} \\
v_y(x+\Delta x, y+\Delta y, z+\Delta z, t) &= v_y(x,y,z,t) + c_{\epsilon_y}\, \epsilon_y(x+\Delta x, y+\Delta y, t)^{a_h} \Delta l^{H_h} + c_{\theta_y}\, \theta_y(z+\Delta z, t)^{a_v} \Delta z^{H_v} \\
v_z(x+\Delta x, y+\Delta y, z+\Delta z, t) &= v_z(x,y,z,t) + c_{\epsilon_z}\, \epsilon_z(x+\Delta x, y+\Delta y, t)^{a_h} \Delta l^{H_h} + c_{\theta_z}\, \theta_z(z+\Delta z, t)^{a_v} \Delta z^{H_v}
\end{aligned} \tag{18}
$$

where $\Delta l = \sqrt{\Delta x^2 + \Delta y^2}$ is the (classical), horizontal distance from the anenometer and $\theta$ is the buoyancy force flux but it is supposed independent of $\varepsilon$ contrary to $\phi$ (Eq.17 ). Unfortunately, this model suffers from a number of basic problems:

– equalities in distribution are replaced by deterministic equalities, which oversimplify and trivialise the dynamics for each realisation

– the flux density $\varepsilon$ is defined only along 3 directions instead along all directions (Eq.11), whose corresponding components $(\varepsilon_x, \varepsilon_y, \varepsilon_z)$ are supposed to be independent, which is not a tenable assumption

– the introduction of $\theta$ independent of $\varepsilon$ is purely ad-hoc to additively (!) introduce a second isotropic scaling (!), therefore the change of scaling is reduced to a linear cross-over

– the scale of the flux densities $\varepsilon$ and $\theta$ are implicitly taken as the scale of the simulation resolution instead of the pair separation scale(Eq.15). The resulting mismatch between both scales will introduce a statistical bias

– moreover the density values are arbitrary taken at the locations $(x+\Delta x, y+\Delta y$ for $\varepsilon)$ and $z+\Delta z$ for $\theta$. This introduces a dissymmetry that is not without consequences.

– all time steps are fully independent with the exception of the empirical velocity values $v(x,y,z,t)$ measured by the 3D sonic anemometer

The limitations of this model are therefore extremely strong. Many of them would have been resolved with the help of scalar anisotropic cascades, recalled above (see previous paragraph). But to fully overcome them would require to consider their extension to vector fields (Schertzer and Tchiguirinskaia, 2015, 2020). This remains outside the scope of this paper and this simplistic approach is used for a sort of exploratory step focused on the interaction of drops and a given velocity field. The obtained results have therefore to be critically examined, since many of them are sensitive to the aforementioned oversimplifications.

### 4.1.3 Practical implementation

The fields $\epsilon$ (i.e. the ones for the horizontal shift) are simulated in space time with a size 729 x 729 x 64 using discrete UM cascades and Eq. 10 to obtain either positive or negative values. The 729 refer to space while the 64 refer to time. Before stating the physical resolution of the field, it should be explained that a simple anisotropy between space and time is accounted through a scaling anisotropy coefficient $H_t$. In such framework, when the spatial scale of the data is changed by a ratio of $\lambda_{xy}$, then the temporal scale should be changed by a factor of $\lambda_t = \lambda_{xy}^{H_t}$. $H_t$ is expected to be equal to $1/3$ (Marsan et al., 1996), hence when the spatial scale is multiplied by 3, the temporal scale should be multiplied by 2 (i.e. $3^{1-1/3} \approx 2.08$) (Biaou et al., 2005; Gires et al., 2014). These fields are assumed to cover an area of size 40 $km$ x 40 $km$ x 1024 $s$, which is needed for drift of 0.1 $mm$ drops during their fall when wind is strong. It means that a voxel is of size is 53 $m$ x 53 $m$ x 16 $s$.

The fields $\theta$ (i.e. the ones for the vertical shift) are of size 512 x 64 covering a physical are of 1600 $m$ x 1024 $s$, meaning that a pixel size 3 $m$ x 16 $s$. All UM fields are simulated with $\alpha = 1.7$, $C_1 = 0.2$ as in the previous section.

Finally, at any point (x,y,z,t) a bi or tri-linear interpolation is implemented to obtain the value of the field from the nearest points. The value of the prefactor were set to $c_{\epsilon_x} = c_{\epsilon_y} = 0.3$, $c_{\epsilon_z} = 0.1$ and $c_{\theta_x} = c_{\theta_y} = c_{\theta_z} = 0.01$, through an heuristic approach of trial and error to get some realistic fluctuations. In the future, it would obviously be needed to tune them to local wind properties. However such tuning is outside the scope of this section, which aims more at being a proof of concept.

### 4.2 Illustration

In order to illustrate the suggested process, let us consider a 0.5 $mm$ drop during the low wind event. Its initial position is (0,0,1500) in $m$. It is 'dropped' with no horizontal velocity and a vertical one equal to its terminal fall one. The anemometer is assumed to be located at at (0,0,100) $m$. Then Eq. 5 is implemented. At each time step the local wind is assessed using the methodology described in the previous paragraphs.

The actual total wind perceived by the drop (i.e. input in Eq. 5) is recorded and displayed in last row of Fig. 7. It corresponds to the sum of the wind from the anemometer (first row in Fig. 7) plus a wind shift field (middle row in Fig. 7). This yields to a given trajectory in space which is shown in Fig. 8. Projected trajectory on the plans (x,y) and (x,z) are also shown. This trajectory exhibits a non linear complex pattern which results from the turbulent nature of the wind. It should be mentioned that

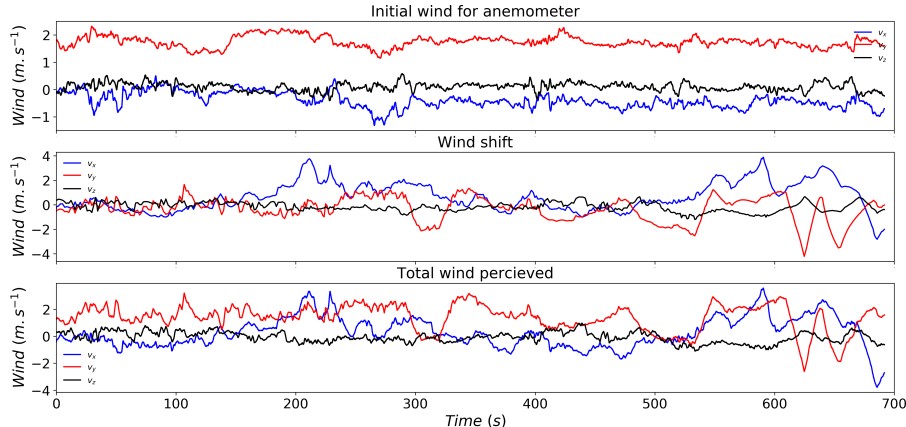

**Figure 7.** Temporal evolution with 0.01 $s$ time steps of the wind data from 3D sonic anemometer (a), the wind shift (b) and the total wind perceived (c) by the 0.5 $mm$ drop falling from the position (0,0,1500) during the low wind event. (c) is actually the wind input used to obtain the trajectory of Fig. 8.

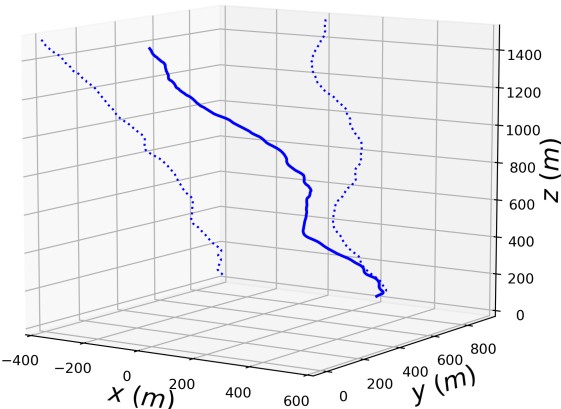

**Figure 8.** Trajectory (solid line) of a 0.5 $mm$ drop in a turbulent wind field for the low wind event. The dotted lines correspond to the trajectory projected on the (x,z) and (y,z) plan

not accounting for oblateness affects not only the vertical velocity but the whole trajectory as well. Depending on the model (spheres or oblate drops), shifts of more than few hundreds meters are found even for large drops for some events.

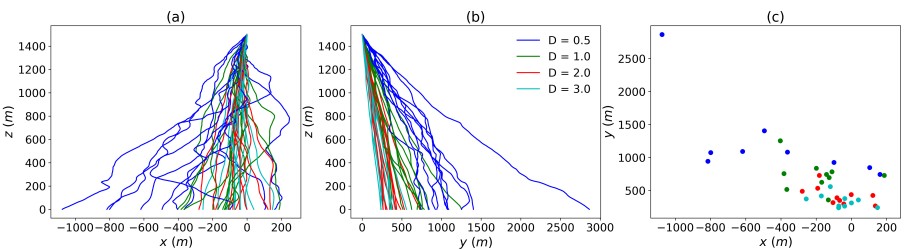

**Figure 9.** (a) and (b) Trajectories of drops of various sizes falling from the (0,0,1500) position, projected on the (x,z) and (y,z) plan respectively. Low wind event is used. The 10 curves correspond to different realizations of the wind shift field. (c) Position of the ground impact of the various drops

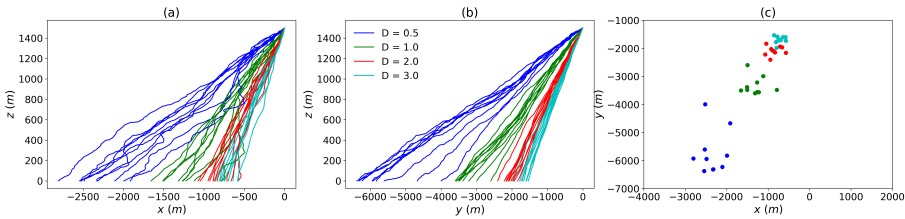

**Figure 10.** Same as in Fig. 9 but for the strong wind event

### 4.3 Sensitivity to the wind shift field

The process to generate an estimation of a 3D wind field is actually stochastic through the UM fields $\epsilon$ and $\theta$ used in Eq. 18. In this section, the sensitivity to the given realization of the process is discussed. In order to achieve that, 10 wind samples are generated with the same input parameters (described in section 4.1) and the corresponding trajectories for drops of size 0.5, 1, 2 and 3 $mm$ are computed.

For the low wind event, the projected trajectories are displayed in Fig. 9.a and b. The position of the drop when they reach the ground is in Fig. 9.c. The spread of the drops strongly depends on their size, with a decrease as drop size increases. Indeed $\Delta x$ ($x_{max} - x_{min}$) is equal to 1238, 591, 415 and 404 $m$ for drop of size 0.5, 1, 2 and 3 $mm$ respectively. For $\Delta y$ the values are 2123, 897, 461 and 322 $m$. Such decrease is due to a combination of the fact that smaller drops are more subject to wind fluctuations (Section 3), and that they spend more time in the atmosphere (section 2) before they reach the ground. Similar trends are retrieved for the strong wind event (Fig. 10) with a stronger absolute shift. In that case the value for $\Delta x$ are 890, 868, 500 and 448 $m$ respectively. For $\Delta y$ they are 2382, 1003, 568 and 448 $m$.

### 4.4 Illustration of impact on rainfall retrieval with weather radars

In this last section, initial investigations toward understanding the consequence of previous work on quantitative rainfall measurement with weather radars are carried out. Indeed, weather radar measure rainfall at a given altitude while hydro-

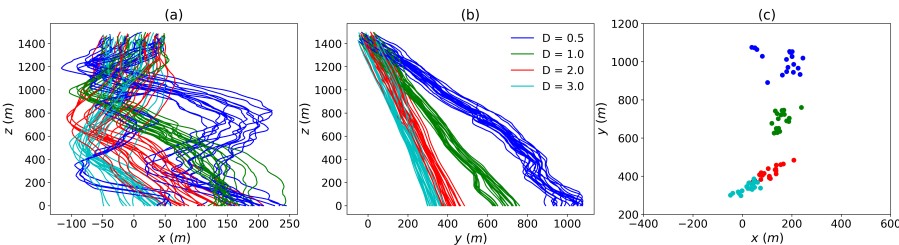

**Figure 11.** (a) and (b) Trajectories of drops of various sizes falling from a 100 m cubic voxel centered on (0,0,1450) position, projected on the (x,z) and (y,z) plan respectively. Low wind event is used. For each size, the 20 drops are dropped every 15 s (hence over a total duration of 5 min). A single realization of the wind shift field is used. (c) Position of the ground impact of the various drops

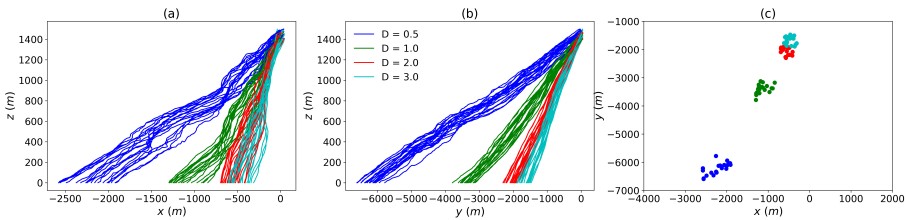

**Figure 12.** Same as in Fig. 11 but for the strong wind event

meteorologist are interested about rainfall at ground level. During their fall, significant shift can occur. In order to study it, the following process is implemented. During five minutes, one rainfall drop is dropped every 15 $s$ from a random position within a voxel of size 100 $m$ centered on (0,0,1450) $m$. Hence it covers a total duration of 5 $min$. The trajectories and positions on the ground of the drops is then studied. This enables to basically mimic the measurement of a weather radar at its typical gate size and temporal resolution.

Fig. 11 displays the trajectories and ground impact location in the case of the low wind event for a given realisation of the stochastic wind shift. A shift of more than 1 $km$ for small drops is found and more than 300 $m$ for 3 $mm$ drops. As noted in the previous section, the spread of drops at ground level tends to decrease with increasing drop size. Indeed $\Delta x \, (x_{max} - x_{min})$ is equal to 206, 119, 165 and 121 $m$ for drop of size 0.5, 1, 2 and 3 $mm$ respectively. For $\Delta y$ the values are 184, 134, 112 and 88 $m$. For the strong wind event (Fig. 12), shifts of more than 6 $km$ and 1.5 $km$ are reported for drops of size 0.5 $mm$ and 3 $mm$ are retrieved. Similar results as for the low wind event are found with regards to the spread. The corresponding figures are of 665, 450, 284 and 307 $m$ for $\Delta x$, and of 819, 664, 429 and 420 $m$ for $\Delta y$.

As previously pointed out, this spread is due to the fact that smaller drop spend more time in the atmosphere and are more sensitive to wind fluctuations. Indeed the duration of fall from 1500 $m$ to the ground at 0 $m$ is equal to 716, 378, 238 and 192 $s$ for drops of size 0.5, 1, 2 and 3 $mm$ respectively. Given that high resolution radar pixels are typically of size few hundred meters, one should note that drops within a given voxel at measurement height can reach ground within an area of size 3 $km$ x

6 $km$. Even within a drop diameters class, shifts are of few radar pixels. Given that drop size distribution also varies, such shift

can significantly affect rainfall retrieval, even in low wind conditions.

## 5   Conclusions

In this paper we aimed at better understanding the behaviour of individual rainfall drops falling from typically 1500 $m$. In a first step we developed a new approach to compute the drag coefficient accounting for drop oblateness and findings in fluid mechanics. This was validated for drop of equivolumic size of up to 4 $mm$ through the comparison between retrieved terminal

fall velocity and commonly used formula.

Then the temporal evolution of horizontal drop velocity under turbulent wind constraints was studied. It appears that multifractal features of the input wind are simply transferred to drop velocity with an additional fractional integration and slight time shift. UM parameter $\alpha$ and $C_1$ are basically conserved while $H$ is increased. The increase ranges from 0.1 for 0.1 $mm$ size drops to 0.8 for 1-1.5 $mm$ size drops. It remains rather constant for larger drops.

Finally the trajectories of drop of various size falling form 1500 $m$ was studied as a proof of concept. For this, 100 Hz anemometer data was used and an approach to simulate realistic fluctuations of wind in space was developed. It notably enables to analyse how drop are shifting during their fall between their location measurement by weather radars and ground impact. For a strong wind event, drops located within a radar gate in altitude during 5 $min$ are spread on the ground over an area of size few kilometers. Spread for drops of a given diameter are found to cover few radar pixels.

In order to explore further the consequences of these findings on quantitative rainfall estimation with weather radars, further investigations are needed. More precisely (i) the model to simulate wind fluctuations should be improved, notably to use vector simulations and tune the prefactors according to local wind conditions; (ii) space-time outputs of numerical weather prediction models could also be tested to retrieve wind fields; (iii) actual drop size distribution should be used to assess better the impact for ground estimation of precipitation, which implies making some simulations for a much larger number of drops. ; (v) longer

period of time should be tested to investigate where the water volume (i.e. all the drops) of a given radar gate fell during an event. For the two last points, data is available within the RW-Turb project. Such step would then need to be repeated over various radar gates to derive updated radar maps. Given limited computation power that will not allow to simulate the trajectories of all the drops, some statistical behaviour according to each radar gate and wind conditions would need to be designed and then computed. It should also be stressed that only individual drop are currently handled. This means that the

developed methodology does not account neither for collision, aggregation between drops, nor for breakup. Such processes are also known to affect drop velocities by changing their size and shape. Future investigations should also aim at accounting for them. Finally, it should also be stressed that the method developed stochastically simulates wind fluctuations at small scales. It means that the output will not be a deterministic radar measurement but an ensemble of possible realistic outputs, out of which a probability distribution could be derived. Such probabilistic approach is discussed in Kirstetter et al. with a focus on intrinsic

radar uncertainties and not wind drift.

*Acknowledgements.* The authors gratefully acknowledge partial financial support from the Chair "Hydrology for Resilient Cities" (endowed by Veolia) of Ecole des Ponts ParisTech, the Île-de-France region RadX@IdF Project, and the ANR JCJC RW-Turb project (ANR-19-CE05-0022-01)

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
