# Peer review of "3D trajectories and velocities of rainfall drops in a multifractal turbulent wind field"

_Atmospheric Measurement Techniques, 2021_

## Author Comment (AC1)

Authors would like to thank the reviewer for its careful reading, comments and inspiring suggestions. We took them into account in the revised. Hopefully, you will be satisfied with it. Please find below a point by point answer to your comments.

The paper models the 3D trajectories and velocities of discrete raindrops accounting for corrections for raindrop oblateness. The paper is original and results are sound, but there are some comments that need clarification before the paper is accepted for publication.

Section 2.1 needs improvement mainly on the definition/description of some of the variables/equations. For instance, the wind is a vector and has 3 components, but the definition of v_wind does not say in which direction (is it in the 'z' direction?).

It was clarified that v_wind is indeed a vector with three components.

What's "SA"? how did you come up with the equations shown in Lines 105 and lines 114 for MPA.

"SA" corresponds to surface area. This was clarified after this first use. Equations were obtained via integral calculus. Please find below a figure to illustrate the computations. This was added as well in the paper to clarify this point.

[Figure]

The force balance (FW = FB+ FD) is valid with the assumption that the particle reached terminal velocity and therefore any additional force due to acceleration is zero. Therefore, if the particle reached terminal velocity, should not dv/dt be equal to zero? However, Eq 1 shows that dv/dt is not zero. Could you please clarify?

In this paper, we are interested in the motion of a particle within a changing wind, meaning that it never reaches its terminal fall velocity. So we are simply using Newton's second law which equals the mass times the acceleration to the net force (here it was divided by the mass). This was clarified in the manuscript.

In addition, the force balance gives:

$F_w = F_B + F_D$

where $F_w$ is the weight of particle, $F_B$ buoyancy and $F_D$ the drag. This equation leads to a well-known expression in fluid mechanics for the terminal velocity of a single particle given by:

$v^2 = 4*D*g/3/C_D* (\rho_p - \rho_{air})/\rho_{air}$

where $\rho_p$ and $\rho_{air}$ is the density of the particle and air respectively, $C_D$ the drag coefficient, g gravity, D particle's diameter. So if the particle's velocity v is equal to $v_{rel} = v_{wind} - v_p$ in your notation (assuming $v_{wind}$ is in the z direction), then any change in $v_{wind}$ over time will affect $v_{rel}$ and $C_D$ ($C_D$ is a function of Re and Re a function of $v_{rel}$). So it is unclear how you came up with your Equation 1 without including the time derivatives of $C_D$ and $v_{wind}$. Perhaps I misunderstood something, but if you can elaborate please.

Authors would like to thank the reviewer whose comment help spot a typo in Eq. 1. Indeed a division by rho_water was missing in the term corresponding to the drag force. This has been corrected. It did not affect the scripts which were correct. The equation you are mentioning for terminal fall velocity is obviously retrieved as well. Due to copy/paste of latex equation, it was also reflected in Eq. 5 which has now been corrected as well.

The notation regarding v_wind which is a vector have been clarified.

All the above does not account for raindrop breakup or aggregation and only applies for discrete particles that do not interact with each other. However, we know this is an important process in precipitation and this will affect v through the increase/decrease of D. Given the fact that you are using a more complex model to work out $C_D$ and account for raindrop oblateness, what are the implications of breakup/aggregation in your results?

The reviewer is correct that these processes are currently not dealt with in this paper which handles individual drops. Following your comment, this has been clarified and is now mentioned as a limitation in the conclusion.

Section 3.1. Recommendation: to use a different variable for C1 in Eq 7 to avoid confusion with 'c1' in Eq 3.

Thanks you for spotting this issue. Given that C1 in Eq. 7 is a very common notation, authors changed the "c"'s in Eq. 3 to "a"'s to avoid any confusion.

Section 3.2 The real part 'Re' might be confused with the Reynolds number 'Re'.

It was changed to "Real" to avoid any confusion and improve clarity.

Eq 11. be consistent with the variable definition. is ux, uy and uz the same as vx, vy and vz in Figs 5 and 6?

It is corrected in Eq. 11 to ensure consistent notations.

From the conclusions, it is clear that wind effects are important especially during strong winds. However, it is unclear how to correct radar rainfall estimates on the ground to account for this. Perhaps the authors can elaborate further.

As suggested by the reviewer, the conclusion has been updated to elaborate more on this issue.

From a practical point of view, what's the difference in trajectories/displacements shown by this model and the one that assumes spherical particles (when computing CD). Is the additional complexity in the modelling adding any value? I would like to see the differences in terms of displacements as well.

There are indeed some differences between trajectories (and therefore ground position) with or without accounting for oblateness. Please find below some examples of trajectories (projected on vertical plans) for drops of various sizes and for the two events. There are some differences of more than few hundred meters even for large drops. A sentence was added to mention this in section 4.2. For now this kind of figures was not inserted, but if the reviewer feels that it should be because the comment is not enough, this can obviously be done.

Low wind

Deq = 2 mm

[Figure]

Deq = 3 mm

[Figure]

Deq = 4 mm

Strong wind

Deq = 2 mm

[Figure]

Deq = 3 mm

Deq = 4 mm

[Figure]

Spelling mistakes:

"equivolumic" (line 193), "withing a voxel" (line 272).

What was the issue with "equivolumic" ? Miss spelled "within" was corrected.

---

## Author Comment (AC2)

Authors would like to thank the reviewer for its careful reading, comments and inspiring suggestions. We took them into account in the revised. Hopefully, you will be satisfied with it. Please find below a point by point answer to your comments.

The manuscript by Gires et al., developed an approach to simulate the 3D trajectory of raindrops by considering the fluctuation of small-scale wind in both space and time and drift patterns of non-spherical raindrops. This study overcomes the limitations of traditional coarse wind data and effectively corrects the rainfall observation under wind drift effect. The objectives of the study clearly explained in the introduction section, as well as the contribution of work. However, some figures in the article lack elaboration, the text also has obvious formatting problems. My detailed comments are provided below:

Despite some minor issues raised below, I think the paper would be a useful addition to the literature and recommend publication after the authors address the comments/questions below.

Thanks for your positive feedback !

It's not clear what is the difference between Í $v_{rel}$ and $v_{rel}$? I cannot find any explanation about $v_{rel}$ (v without underscore) in your manuscript. However, both appear frequently in subsequent equations (e.g. line 76, line 80, line135).

\underline{v}_{rel} with an underline is the vector with three components while {v}_{rel} without any underline is the euclidean norm of this vector. This was clarified in section 2.1.

As for wind field generation part, there are still a few confusions. In line 222, what is the standard for distinguishing high and low wind speed? Is it the average speed of the three directions or something else?

The distinction was based on the average total horizontal wind (i.e. 1.8 vs. 11.8 m/s). The term high and low are arbitrary and simply refer to the difference between the two events. This was clarified in section 4.1.

In line 232, what is the resolution of the 729 x 729 x 64 grid? Besides, what does 64 refer to? Time or altitude?

The 64 corresponds to time. The resolution is actually stated at the end of the paragraph. A sentence was added to help the reader understand more easily.

The manuscript lack of the description of study area and the 3D sonic anemometer instrument. Despite the author's citation of the data, I would still like to know how the instrument works and how the wind speed data is organized. Please improve this section by adding more detailed information.

The anemometer is installed at 78 $m$ on meteorological mast located on the Pays d'Othe wind farm in the framework of the ANR RW-Turb project. The wind farm is at roughly 120 $km$ South-East of Paris on a slightly sloppy area. More details can be found in the data paper under discussion at ESSD (\citet{essd-2021-463} and in the data set (\citet{gires_auguste_2021_5801900}).

Some details were added and now that the corresponding data paper is online for discussion at ESSD EGU journal, the full reference is cited for interested reader.

Besides, I cannot find any description of Fig. 6 in the manuscript

Wrong name for figure reference was used in beginning of section 4.2. This is now corrected. Thanks for spotting this !

Below are specific comments and suggestions:

Line 25: "km" in the brackets may be italicized.

Done

Line 36-37: "for example" appears consecutively.

This was corrected.

Line 73/79/87/104...: Please remove spaces before colon characters.

This was corrected.

Line 96: Please add a period at the end of the sentence.

This was corrected.

Figure 1c: What is the difference between abscissa and ordinate? Which is the retrieved one?

D in abscissa is the diameter used in Thurai's formula (Eq. 3). From this, a shape is found and a volume can be computed. From this volume, an equivolumic diameter can be assessed. It is this equivolumic diameter that is displayed in the vertical axis. It is obviously quite close ! The aim is simply to show how the small discrepancies are accounted for. This was clarified in the manuscript.

Figure 1d: Please add y-label.

This was done in the updated version of the figure.

Figure 1f: Please change "(e) Terminal fall velocity vs. equivolumic diameter" to "(f) Terminal fall velocity vs. equivolumic diameter".

This was corrected.

Line 107: Does "SA" refer to surface area?

Yes, and this is now mentioned.

Line 119: I wonder if the drag coefficient is $C_d$, $C_D$ or $c_D$ (in Eq. 4, Fig. 1e and line 76)?

c_D is now used everywhere. Thank you for spotting this issue.

Line 140: "â     " may be "â     t".

Indeed a "t" was missing and it is now corrected.

Figure 2: Is D different from $D_{eq}$? Please clarify.

Yes, and this is now clarified in the caption.

Line 198: Do you refer to 0.1 mm?

Yes, this was corrected.

Line 200: "For the 2 mm drop, the scaling is slightly degraded but remains good (r2 = 0.95 for q = 1.5). α = 1.69, C1 = 0.14 and H = 0.79 is found." I prefer to combine the two sentences into one.

The sentence was rephrased.

Figure 4: Please check the numbering order.

The figure and caption seem to be correct. What are you referring to ?

Line 221/225: Please check format of the citations.

This was corrected.

Line 242: Do you mean "at any point (x,y,z,t), a bi- or tri-linear interpolation…"ï¼

Yes and the parenthesis were added.

Line 258: Please list the specific parameter information of these 10 wind samples.

It is the same input parameters for all the samples and the parameters are listed in section 4.1. This is now clarified.

Line 275: Is the wind shift field here one of the 10 types of wind samples used in section 4.3? If yes, it is recommended to describe and highlight in Fig. 8 (e.g. in dot line); same recommend in Fig. 11.

Given that the figure is already  slightly messy, authors are not sure that adding this information will be helpful. If you consider this is really important, this could indeed be envisaged.

Line 281: Please add units to the Δy numbers

This was done.

Line 289: Please unify the number format, such as "1 500" or "1500"?

1500 is now used everywhere.

Line 295: "The increase ranges from 0.1 for 0.1 mm size drop to 0.8 for drops of size 1-1.5 mm." change to "The increase ranges from 0.1 for 0.1 mm size drops to 0.8 for 1-1.5 mm size drops."

This was done.

---

## Author Response (AR2)

Please find below a point by point response to reviews. Hopefully you will be satisfied with them.

Report #1

The authors have addressed all the issues raised in my previous review and I believe the paper is suitable for publication.

Thanks for your positive feedback.

Minor comments:
Copyright statement is missing.

Authors believe that this will be handled at the editorial stage. For now, we simply removed it as suggested on the manuscript tracking tool.

Do you mean "equivolumetric' (instead of 'equivolumic')? " I guess both are valid.

Yes, and authors believe that both are valid.

Report #2

Thank you very much for the authors clearly responding and/or revising my previous comments. Currently I have no more corrections related to the manuscript.

Thanks for your positive comment !

However, I still suggest the authors to add one paragraph of discussion about how the proposed method can be used in the radar rainfall estimation and the possible challenges. In addition, Yang et al. (2020) proposed a data-based adjusted method for wind effects on radar rainfall estimations. It will be useful for the community if the authors can discuss their findings with it.

Yang, Q., Dai, Q., Han, D., Zhu, Z., & Zhang, S., 2020, Uncertainty analysis of radar rainfall estimates induced by atmospheric conditions using long short-term memory networks, Journal of Hydrology, 590: 125482

The last paragraph of the conclusion was actually strongly updated following the previous comment to discuss more this issue. Nevertheless, an additional comment highlighting the intrinsic stochastic nature of small scale wind fluctuation simulation and its consequences on radar measurement was add in this new version.

With regards to the paper mentioned, it is indeed relevant and is now cited in the introduction along with already cited papers relying on model outputs to account for wind drift.